# Integrating Single-Cell RNA-Seq and ATAC-Seq Analysis Reveals Uterine Cell Heterogeneity and Regulatory Networks Linked to Pimpled Eggs in Chickens

**DOI:** 10.3390/ijms252413431

**Published:** 2024-12-15

**Authors:** Wenqiang Li, Xueying Ma, Xiaomin Li, Xuguang Zhang, Yifei Sun, Chao Ning, Qin Zhang, Dan Wang, Hui Tang

**Affiliations:** Shandong Provincial Key Laboratory for Livestock Germplasm Innovation & Utilization, College of Animal Science and Technology, Shandong Agricultural University, 61 Daizong Street, Taian 271018, China; 2021010079@sdau.edu.cn (W.L.); maxueying1997@163.com (X.M.); 17662581638@163.com (X.L.); 15589250292@163.com (X.Z.); bjyx1823095@163.com (Y.S.); ningchao@sdau.edu.cn (C.N.); qzhang@sdau.edu.cn (Q.Z.)

**Keywords:** pimpled egg, chicken, scRNA-seq, scATAC-seq, uterus

## Abstract

Pimpled eggs have defective shells, which severely impacts hatching rates and transportation safety. In this study, we constructed single-cell resolution transcriptomic and chromatin accessibility maps from uterine tissues of chickens using single-cell RNA sequencing (scRNA-seq) and single-cell ATAC sequencing (scATAC-seq). We identified 11 major cell types and characterized their marker genes, along with specific transcription factors (TFs) that determine cell fate. CellChat analysis showed that fibroblasts had the most extensive intercellular communication network and that the chickens laying pimpled eggs had amplified immune-related signaling pathways. Differential expression and enrichment analyses indicated that inflammation in pimpled egg-laying chickens may lead to disruptions in their circadian rhythm and changes in the expression of ion transport-related genes, which negatively impacts eggshell quality. We then integrated TF analysis to construct a regulatory network involving TF–target gene–Gene Ontology associations related to pimpled eggs. We found that the transcription factors *ATF3*, *ATF4*, *JUN*, and *FOS* regulate uterine activities upstream, while the downregulation of ion pumps and genes associated with metal ion binding directly promotes the formation of pimpled eggs. Finally, by integrating the results of scRNA-seq and scATAC-seq, we identified a rare cell type—ionocytes. Our study constructed single-cell resolution transcriptomic and chromatin accessibility maps of chicken uterine tissue and explored the molecular regulatory mechanisms underlying pimpled egg formation. Our findings provide deeper insights into the structure and function of the chicken uterus, as well as the molecular mechanisms of eggshell formation.

## 1. Introduction

Eggshell formation is a complex process in the avian uterus that involves multiple ion transport proteins [1]. It lasts about 20 h, during which specific interactions between ions and matrix proteins in the uterine fluid determine the ultrastructure and strength of the eggshell [2]. Breakage-induced losses caused by poor shell quality account for 12–20% of total egg yield per year [3]. Therefore, studying the gene expression and chromatin accessibility of chicken uterine tissues is crucial for elucidating the molecular mechanisms of eggshell formation.

Pimpled eggs have calcification granules on their surface, which not only make them more prone to breakage but also deter consumers [4,5]. The mechanisms of pimpled egg formation have been investigated before at the RNA, protein, and metabolite levels [6]. For example, Song et al. used tandem mass tag (TMT)-based quantitative proteomics to discover that CALM1, SCNN1G, and PHB1 proteins are involved in pimpled egg formation through calcium signaling pathways and immune response regulation [7]. Also, metabolomic analysis of uterine fluid from chickens laying eggs of varying shell qualities revealed that phosphatidylcholine, diacylglycerol, and verapamil are critical for eggshell calcification [8]. In our own previous work, we constructed a competing endogenous RNA (ceRNA) regulatory network associated with pimpled egg production using whole-transcriptome sequencing [9]. We identified 15 key genes related to ion transport and immunity, such as *OVAL*, *CABP1*, and *HPCA*, that are involved in regulating eggshell quality. These studies showed that eggshell formation is associated with uterine immune responses and ion transport capabilities [5,9,10]. The number of pimpled eggs, as well as the number of calcium granules on the shells increase, with hens’ age [6]. This is similar to middle-aged mammals, where aging of the uterus can lead to infertility and pregnancy complications [11,12]. The formation of pimpled eggs is closely related to the impairment of normal uterine function in hens. This damage may be caused by uterine inflammation, malnutrition, or a decline in uterine function associated with aging [13,14]. Therefore, comparing the heterogeneity of uterine tissue cells between hens that lay normal eggs and those that lay pimpled eggs can provide valuable insights into the mechanisms underlying pimpled egg formation.

Single-cell RNA sequencing (scRNA-seq) enables the analysis of gene expression profiles at the single-cell level, which can reveal heterogeneity within cell populations and uncover novel rare cell types [15]. For example, Plasschaert et al. conducted single-cell analyses of human bronchial epithelial cells and mouse tracheal epithelial cells, leading to the first identification of the rare cell type “pulmonary ionocytes” [16]. Before this discovery, ionocytes were only known from mammalian kidneys and the skin of African clawed frog (*Xenopus laevis*) tadpoles [17,18]. scRNA-seq is emerging as a key method for a variety of studies, including research on uterine development and associated diseases [19,20]. For example, Wang et al. characterized the transcriptomic transformation of the human endometrium across the entire menstrual cycle at single-cell resolution [21]. They analyzed seven types of endometrial cells, including a previously uncharacterized ciliated cell type, and found that the menstrual cycle is associated with functional changes in epithelial and fibroblast cells [21]. In another study, Punzon-Jimenez et al. compared the cellular atlas of the aging myometrium between peri- and postmenopausal women and discovered that myometrial tissue aging is primarily driven by decreased contractility, increased fibrosis, and heightened inflammation [11]. Thus, scRNA-seq has significantly improved our understanding of the cellular composition and function of the uterus [22]. However, current research mainly focuses on human and mouse models, whereas studies related to poultry, particularly the uterine tissues of laying hens, are largely neglected.

Single-cell assay for transposase-accessible chromatin sequencing (scATAC-seq) allows for the analysis of chromatin accessibility and transcription factor (TF) binding in single cells [23]. In combination with scRNA-seq, it can reveal cell-specific regulatory landscapes and gene expression mechanisms, which can provide deeper insights into intracellular gene regulation and functional heterogeneity [24,25]. Thus, by combining scATAC-seq and scRNA-seq, higher-resolution analysis of uterine cells can be achieved. However, currently, most single-cell studies related to the uterus are limited to scRNA-seq alone [19,20,26]. As an exception, a recent study generated a high-quality cellular atlas composed of four tissue sites from the postmenopausal female reproductive tract by integrating scRNA-seq and scATAC-seq [27]. The integration of these two technologies is crucial for deciphering the cellular complexity of tissues. For example, Lengyel et al. used this approach to identify three subclusters of endothelial cells, which could not have been achieved with either method alone [27]. However, to our knowledge, scATAC-seq has never been used in studies related to avian reproductive organs.

In the present study, we combined scATAC-seq and scRNA-seq to construct the first cellular atlas of the chicken uterus. By integrating transcriptome data with chromatin accessibility profiles, we revealed the heterogeneity of 11 cell types and systematically analyzed their abundance, specific gene expression patterns, cell–cell communication (CCC) networks, and key regulatory factors in cell fate determination. By comparing single-cell transcriptomic and chromatin accessibility differences between hens laying pimpled eggs and those laying normal eggs, we identified genes, TFs, and signaling pathways likely associated with pimpled egg formation. Our findings offered key resources for a better understanding of the structure and function of avian uterine tissue and laid the theoretical foundation for future breeding efforts aimed at improving egg quality. In addition, for the first time, we report the identification of ionocytes in the chicken uterus. This cell type has not been observed in uteri of mammals such as humans, mice, or pigs, suggesting there are differences in uterine cell types between birds and mammals. This discovery may provide new insights into evolutionary differences and functional adaptations of reproductive systems across species.

## 2. Results

### 2.1. scRNA-Seq Identifies Major Cell Types of Chicken Uterine Tissue

To obtain a cell population profile of uterine tissues of laying hens, we examined six hens, of which three laid normal eggs and three laid pimpled eggs. A total of 70,201 cells passed QC and batch correction, with an average of 1892 genes and 5776 unique molecular identifiers (UMIs) detected per cell (Appendix A). A total of 7795 doublets and apoptotic cells were removed according to Seurat filtering criteria. Through dimensional reduction and unsupervised clustering of all 62,406 cells following the UMAP protocol (https://umap-learn.readthedocs.io/en/latest/ (accessed on 25 March 2024)), 21 distinct clusters were identified (Figure 1A). We manually annotated 11 different cell types using typical marker genes from the literature and PanglaoDB: epithelial cells, luminal epithelial cells, ciliated epithelial cells, endothelial cells, fibroblasts, ionocytes, T cells, macrophages, natural killer (NK) cells, B cells, and erythrocytes (Figure 1B, Appendix A).

Based on expression patterns, we generated bubble plots (Figure 1C) and UMAP projections (Appendix A) for marker genes to demonstrate the accuracy of cellular annotation. GO enrichment analysis allowed for us to identify biological processes associated with the specifically expressed genes of each cell type, revealing a strong alignment with the characteristics of the cells (Appendix A). For example, fibroblast populations specifically expressed *DCN*, *COL1A1*, *MGP*, and *OGN*, and GO analysis revealed differential gene enrichment in regions of the extracellular space (*p* < 0.01), focal adhesion (*p* < 0.01), and calcium ion binding (*p* < 0.01). Similarly, the T cell population expressed *GNLY*, *SRGN*, and *RGS1*, and canonical genes such as *CD3D* and *CD3E*, which are involved in the T cell receptor signaling pathway (*p* < 0.01) and positive regulation of T cell activation (*p* < 0.05).

Next, we quantified the differences in cell type proportions across samples and found that trends in cell type distributions were consistent between the PE and NE groups (Figure 1D). We detected the abovementioned cell types in all six samples, with epithelial cell populations consistently representing the largest proportion. Interestingly, there is a significant disparity in the proportions of fibroblast populations between the two groups (Appendix A). Fibroblasts are the primary producers of the extracellular matrix (ECM), supplying matrix proteins and minerals during eggshell formation. The reduction in the proportion of fibroblast populations in the PE group suggests that fibroblasts play a key role in the formation of pimpled eggs.

### 2.2. Cellular Heterogeneity and Differentially Expressed Genes Between NE and PE Groups

In comparison between the NE and PE groups, fibroblasts have the highest number of differentially expressed genes (DEGs), while erythrocytes have the lowest (Appendix A), suggesting significant functional disparities in fibroblasts between the two groups. We further observed that in the NE group, certain DEGs are consistently upregulated across most cell types. For example, *SLC34A2* has elevated expression in all cell types, while *GLIS1*, *ITPR2*, *PER2*, *RASD1*, and *RASL10A* are upregulated in 10 different cell types. These genes are associated with ion transport and circadian rhythms. To better discriminate distinct functionalities and pathways within the same cell types of the two groups, we performed separate GO and KEGG enrichment analyses on the DEGs of each cell type (Appendix A). Hens are subjected to circadian rhythms during egg laying, and clock genes likely directly regulate the process of egg formation [28]. We found that the upregulated clock genes *PER2*, *ID2*, *EGR1*, and *ID3* in the NE group are enriched in the GO term “circadian regulation of gene expression” (*p* < 0.01) across seven cell types. In the NE group, upregulated genes are enriched in GO terms related to ion transport across various cell types, such as “cellular zinc ion homeostasis”, “metal ion binding”, “hydrogen ion transmembrane transport”, and “bicarbonate transport” (Appendix A, (*p* < 0.05)). Notably, we observed elevated expression of *ITPR2*, *ATP2B2*, *ATP2A2*, *PDE4D*, *ATP1A1*, *ATP1B1*, *ATP6V0A4*, *SLC4A7*, *CFTR*, *PDE4D*, and *ANXA2* in fibroblasts of the NE group. These genes are enriched in the GO terms “calcium ion transmembrane transport” (*p* < 0.01), “calcium-transporting ATPase activity” (*p* < 0.01), “sodium ion export from cell (*p* < 0.01)”, and “cellular sodium ion homeostasis (*p* < 0.05)”, indicating their potential association with ECM secretion in fibroblasts (Appendix A). This further highlights that the heterogeneity of fibroblasts between the two groups may be crucial for the formation of pimpled eggs. In the PE group, we observed elevated expression of immune-related genes such as *IGLL1*, *MHCY11*, *B2M*, *BF1*, *CCL26*, *IL15*, *CD74*, *BLB2*, *CD3D*, and *CD3E* (Appendix A). Furthermore, enrichment analysis of upregulated genes revealed the immune-related GO terms “immune response” (*p* < 0.01), “positive regulation of T cell activation (*p* < 0.05)”, “positive regulation of immune response (*p* < 0.05)”, and “T cell receptor binding” (*p* < 0.05, Appendix A). This suggests that the uteri of the PE group hens may be inflamed, which could influence uterine function and promote pimpled egg formation.

### 2.3. Transcription Factor Analysis of Chicken Uterine Tissue

Transcription factors (TFs) are important regulators of gene expression and play a key role in maintaining cellular functionality. To investigate TF activity in hen uterine tissue, we used SCENIC to identify regulatory factors and gene regulatory networks. We identified 280 significant TF regulons, encompassing 7266 genes (Appendix A). The number of genes per regulon ranged from 4 to 4557, with a median of 38. The SCENIC results revealed significant enrichment of *CREB3L1*, *HOXA9*, *MSC*, and *RARB* regulon activity in fibroblasts; *GATA2*, *SOX5*, and *GLIS1* activity in epithelial cells; *TBX3* and *VDR* activity in ductal epithelial cells; *FOXO1* and *TEAD4* activity in endothelial cells; *SOX9* and *DMRT2* activity in ionocytes; *TFDP1* and *MAFG* activity in erythrocytes; *ARID3A* activity in B cells; *RUNX2* and *RUNX3* activity in T cells; and *CEBPA* and *MEF2C* activity in macrophages (Appendix A). Many of these interactions have been experimentally validated, such as *CREB3L1* driving fibroblast differentiation in systemic sclerosis [29], *HOXA9* overexpression in dermal fibroblasts [30], and *GATA2* playing a key regulatory role in bovine rumen epithelial cells [31], which further supports the credibility of our predictions. SCENIC also inferred multiple downstream target genes of TFs. We integrated the results of inter-group differential analysis to identify TFs regulating genes associated with shell characteristics. The TFs *EGR1*, *JUN*, *FOS*, and *ATF4* are upregulated in the NE group and regulate multiple target genes associated with eggshell traits. Additionally, *EGR1* is involved in circadian regulation of gene expression, while *JUN*, *FOS*, and *ATF4* are involved in the MAPK signaling pathway. Significantly, SCENIC predicted 48 differentially expressed target genes of *ATF3*, among which 45 are enriched in GO terms related to pimpled egg traits. Furthermore, *ATF3* expression is upregulated in multiple cell types including fibroblasts, epithelial cells, and luminal epithelial cells in the NE group, suggesting an important regulatory role of *ATF3* for pimpled egg characteristics. Finally, we constructed a regulatory network based on the inferred target relationships, depicting differential transcription factors, differential target genes, and the associated GO terms (Figure 2).

### 2.4. Cell–Cell Communications

To further understand how cells in the chicken uterus collaborate in eggshell formation, we used CellChat to assess intercellular interactions. First, we constructed comprehensive networks of interactions, identifying 735 in the NE group (Figure 3A) and 1419 in the PE group (Figure 3B). Then we classified ligand–receptor pairs from 11 cell clusters into notable signaling pathways, including COLLAGEN, LAMININ, FN1, WNT, APP, and THBS (Appendix A). Hoffmann et al. have identified collagen as a key ECM protein with anti-inflammatory and antioxidant properties [32]. In both the NE and PE groups, fibroblasts and endothelial cells are the primary sources of the COLLAGEN signaling pathway, with fibroblasts contributing more substantially (Figure 3C,D).

We found that the number and strength of interactions among fibroblasts were significantly higher in both study groups compared to cells (Appendix A). In both groups, fibroblasts were the predominant signal transmitters, which suggests that they play a key regulatory role in uterine tissues. In the NE group, ciliated epithelial cells had the highest number of signaling pathways, followed by epithelial cells and endothelial cells. In contrast, in the PE group, fibroblasts had the most signaling pathways, followed by NK cells and epithelial cells. These findings underscore the association between functional changes in fibroblasts and shell formation. To further explore signaling changes in ligand–receptor pathways, we compared the specific pathways uniquely present in each group (Figure 3E). In the PE group, there are 17 specific signaling pathways, including EPHB, FGF, IGF, VEGF, TGF-β, AGRN, and HSPG. In contrast, the NE group only has CD39 and NCAM. CD39 is involved in suppressing T cell function and NK cell cytotoxicity, while NCAM plays a role in the expansion of T cells, B cells, and NK cells. We found that most signaling pathways differentially expressed between the two groups are related to immune functions. At the same time, the altered pathways significantly affect a range of critical biological functions: ECM (e.g., TGFb, HSPG, PERIOSTIN, ANGPTL), angiogenesis (e.g., EPHB, VEGF, PDGF, PROS), immune and inflammatory processes (e.g., CD39, NCAM, SEMA5, SEMA7, CCL, CSF, EDA), and the nervous system (e.g., AGRN and NEGR). These results suggest that changes in fibroblast function lead to modifications of several signaling pathways, including those related to immune response, which ultimately promotes the formation of pimpled eggs.

### 2.5. scATAC-Seq Reveals Chromatin Accessibility Landscapes of the Chicken Uterus

In scRNA-seq, the detection rate of lowly expressed genes, including TFs, is limited. However, understanding TFs and their regulatory activities is essential for deciphering complex biological processes. To further explore regulatory events in chicken uterine tissue, we used scATAC-seq to construct chromatin accessibility profiles for a total of 10,405 cells from the PE (6089) and NE (4316) groups. First, we used Cell Ranger ATAC for QC of fastq data, peak identification, and cell filtering. After preprocessing, on average ~56,244 fragments per cell were detected in sample N1, with 59.2% of fragments mapping to peaks (Appendix A). In sample P1, on average, ~64,577 fragments per cell were detected, with 61.0% of fragments mapping to peaks (Appendix A). To characterize the different genomic elements captured by scATAC-seq, we used ChIPseeker to annotate and statistically analyze the distribution of peaks across various functional regions (promoter, 5′UTR, 3′UTR, exon, intron, downstream, and intergenic) (Figure 4A). The proportions of genomic elements between PE and NE samples exhibited minor differences, with introns comprising the highest proportion at ~40%, followed by promoters at ~35%. Utilizing a clustering algorithm optimized with SNN modularity, we identified 22 and 11 distinct differentially accessible peak clusters in N1 and P1, respectively (Appendix A). Due to almost all cells from sample N1 clustering together, which hindered effective clustering, subsequent analyses primarily focused on sample P1. Using Seurat, we utilized our annotated scRNA-seq dataset to predict cell types in the scATAC-seq data (Figure 4B). The frequency of assigned cell types in the scATAC-seq data roughly matched those of the scRNA-seq data, suggesting a broad correlation between chromatin accessibility and transcriptomic profiles (Figure 4C). To validate the assignment of the scATAC-seq cells, we examined the accessibility of lineage-specific TF motifs within each annotated cell type. Consistent with the predicted annotations, motifs of marker genes exhibit the highest accessibility within their respective cell types (Figure 4D, Appendix A). For example, *COL1A2*, *FBN1*, *COL1A1*, *DCN*, and *COL3A1* are the top five accessible chromatin regions in fibroblast cells, and these are all widely recognized as fibroblast marker genes. Except for *FBN1*, these genes are also highly expressed in fibroblasts according to the scRNA-seq analysis (Appendix A). The increased chromatin accessibility levels of these TFs suggest initiation of regulatory transcriptional mechanisms in fibroblasts. The lack of differential expression detection of *FBN1* may be due to the lower detection rate of lowly expressed genes in scRNA-seq. Overall, these TFs with specific expression are likely candidates for promoting cell fate transitions. In conclusion, the combination of scATAC-seq and scRNA-seq has markedly enhanced our ability to dissect cellular heterogeneity in the chicken uterus.

### 2.6. Joint Annotation of Ionocyte Subtypes Using scATAC-Seq and scRNA-Seq

Plasschaert et al. used scRNA-seq to identify a rare cell type with high expression levels of CFTR, termed ionocytes, which are predominantly located in the airways, renal tubules, and intestines [16,17,33,34]. Ionocytes regulate ion concentrations across cell membranes, maintaining internal ion balance and normal physiological function [35]. To our knowledge, this cell type has not been observed in uterine tissues before. Using scRNA-seq, we identified a cluster in the uterine tissue of hens that potentially corresponds to ionocytes. We examined the expression of a number of marker genes for ionocytes in this cluster and specifically found high expression of *CFTR*, *ATP6V1G3*, *ATP6V0D2*, *ATP6V1C2*, *FOXI1*, *SCG2*, *BSND*, and *PDE1C* (Figure 5A) [16]. We then conducted a GO analysis of these genes, which revealed enrichment in several terms related to ion transport, including “hydrogen ion transmembrane transport”, “bicarbonate transport”, and “chloride channel regulator activity” (Figure 5B). Therefore, we defined this cluster as ionocytes, although this cell type has not been previously reported in uterine tissues of chickens or other animals. We speculate that this might be due to differences between avian and mammalian uterine tissues. After annotating cell clusters in the scATAC-seq data using scRNA-seq, we examined the chromatin accessibility of ionocyte marker genes within the ionocyte cluster. We observed that the ionocyte marker genes *CFTR*, *FOXI1*, *BSND*, *ATP6V1G3*, *ATP6V0D2*, and *PDE1C* have high chromatin accessibility (Appendix A), with *PDE1C* being the most highly enriched motif (Figure 5C). Sato et al. have demonstrated that *PDE1C* can enhance *CFTR* activity in ionocytes [36]. These results further support the accuracy of our annotation. The discovery of rare ionocytes in the chicken uterus is highly important for understanding the heterogeneity of uterine cell types in hens.

## 3. Discussion

Single-cell RNA sequencing (scRNA-seq) and single-cell ATAC sequencing (scATAC-seq) technologies allow for us to observe gene expression changes and chromatin accessibility at the cellular level, facilitating in-depth analysis of cellular heterogeneity and exploration of disease mechanisms [37]. Eggshell formation takes place in the uterus, whose structure and function significantly impact shell quality [6]. Although extensive research on mammalian uteri at the single-cell level has been published, studies on cellular heterogeneity in avian uteri are lacking [38,39].

Pimpled eggs are defective eggs with a higher susceptibility to shell damage and lack of consumer preference, negatively impacting the egg industry [40]. In our study, for the first time, we mapped the transcriptome and chromatin accessibility of chicken uterine tissue at single-cell resolution. Using our novel dataset, we explored the regulatory mechanisms of shell formation in pimpled eggs and the cellular heterogeneity of chicken uteri. We identified a population of ionocytes in chicken uterine tissue, a cell type that has not been found in mammalian uteri. Our findings provide valuable resources for future studies on avian uterine tissue and a deeper understanding of eggshell characteristics.

Based on the literature, the PanglaoDB database, and marker gene expression in the chicken oviduct, we identified 11 main cell types, namely, epithelial cells, luminal epithelial cells, ciliated epithelial cells, endothelial cells, fibroblasts, ionocytes, B cells, T cells, NK cells, macrophages, and erythrocytes. Through functional enrichment analysis, we could further validate the accuracy of these results. For example, genes in fibroblasts were enriched in the functions “extracellular space” and “extracellular matrix” (ECM), consistent with this cell type’s biological functions [41]. We then integrated the results of scRNA-seq and scATAC-seq analyses, revealing a series of cell-specific transcription factors (TFs), which provided deeper insights into cellular transcriptional heterogeneity. For example, we found that *HOXA9* and *PRRX1* have increased chromatin accessibility in fibroblasts, suggesting potential upstream regulatory roles in transcriptional control within these cells [42,43]. *ESRRG* and *FOXP1* have increased chromatin accessibility in epithelial cells [44], while *IKZF1* and *REL* show similar characteristics in T cells [45,46]. As cell type-specific TFs, they regulate the expression of specific genes, which directly influences the functions and characteristics of these cells [47]. Ionocytes play specific physiological roles in fluid regulation at epithelial interfaces and are primarily found in the skin of fish and amphibians and in the human lung and intestine [48,49,50]. They have never been identified in the mammalian uterus, whereas we show that they constitute ~20% of the chicken uterus. We also examined the top enriched motifs within the ionocyte cluster and found high accessibility of ionocyte markers such as *CFTR*, *FOXI1*, and *ATP6V1G3*. In particular, *CFTR*, a strictly regulated anion channel, is typical for ionocytes [50]. Within the annotated ionocytes, *PDE1C* exhibits the highest chromatin accessibility, and *PDE1C* has been shown to enhance *CFTR* expression in ionocytes, with *CFTR* being a classic marker gene of ionocytes [36]. Ionocytes were identified in the chicken uterus but not in the mammalian uterus, which may be due to differences in their reproductive strategies. We speculate that this could be related to the distinct physiological functions of their uteri [51].

Frequency analysis of cells from normal egg-laying (NE) and pimpled egg-laying (PE) groups revealed a significant increase in the proportion of fibroblasts in the NE group. Fibroblasts are involved in the synthesis and degradation of ECM, directly influence the composition of uterine fluid, and play a key role in eggshell formation [52]. Our cell–cell communication analysis corroborated the crucial role of fibroblasts in the avian uterus. In both study groups, fibroblasts are involved in the highest number of signaling pathways and are also the cell type with the highest number of outgoing signaling pathways. Compared to the NE group, the PE group has more signaling pathways. However, it lacks the *CD39* and *NCAM* pathways, which play key roles in regulating immune cell functions [53,54]. By comparative analysis of scRNA-seq data from both groups, we found that upregulated genes in the NE group are associated with ion transport and extracellular space, thus being responsible for provision of minerals and collagen for eggshell formation [6]. Interestingly, “calcium ion transmembrane transport”, “calcium-transporting ATPase activity”, and “intracellular sodium ion homeostasis-related” GO terms were significantly enriched only in the fibroblasts of the NE group. This again underscores the direct impact of fibroblast secretion on uterine fluid composition and eggshell formation. In contrast, genes upregulated in the PE group are related to immune responses, positive regulation of T cell activation, and negative regulation of the fibroblast growth factor receptor signaling pathway. This suggests that the uterine tissue of hens in the PE group may be inflamed, thereby activating a series of immune-related signaling pathways that affect normal fibroblast function, which ultimately leads to pimpled egg formation [10].

The study by Cui et al. found that circadian rhythm genes regulate ovulation and eggshell calcification in the chicken oviduct, thereby playing a key role in eggshell quality [55]. Circadian rhythms control many inflammatory processes, and disruption of the clock can induce or exacerbate inflammation [38,55,56]. On the other hand, inflammation can disturb circadian rhythms, which further amplifies the inflammatory response and exacerbates tissue damage [57]. In line with this, we found that clock genes such as *PER2*, *ID2*, *EGR1*, and *ID3* were downregulated in the PE group uteri, indicating disruption of the circadian rhythms in chicken tissues, which may exacerbate the inflammatory response in chicken uterine tissue [58]. Disruption of the circadian rhythm and inflammation impaired the normal function of the uterine tissue, thereby affecting the process of eggshell calcification. To further investigate the molecular regulatory mechanisms underlying pimpled egg formation, we integrated differential analysis, enrichment analysis, and SCENIC results to construct a TF–target gene–GO term regulatory network. We found that *ATF3*, as an activating TF, targets the most genes in this network and is involved in suppressing inflammatory responses [59,60]. However, downregulation of its expression in the PE group may have exacerbated inflammatory reactions. *JUN* and *FOS* also regulate a large number of target genes. *ATF3* can form homo- or heterodimers with *JUN* or *FOS*, acting either as a repressor or activator of transcription [61,62]. All three TFs are upregulated in the NE group and jointly regulate genes associated with shell formation. For example, the shared target gene *PER2* is rhythmically expressed in the chicken oviduct, affecting Zn and Ca transport [63]. Also, *EZR* is a shared target gene of *ATF3*, *JUN*, and *FOS*, and its expression is downregulated in the PE group, consistent with a previous study [9]. *ATP1A1*, *ATP1B1*, *ATP2B2*, *ATP2A2*, and *ATP6V0A4* are genes responsible for ion transport across cell membranes and maintenance of intracellular and extracellular ion balance [64,65]. These ion pump genes are collectively targeted for regulation by *ATF3* and *EGR1* (except for *ATP6V0A4*). The downregulation of these genes in the PE group affects the concentration of Ca and other ions essential for eggshell formation in chicken uterine fluid, which ultimately leads to the formation of pimpled eggs [1,2]. *EGR1*, as an upstream regulatory TF, not only regulates multiple genes related to ion transport, but is also involved in the circadian rhythm-related regulation of gene expression, metal ion binding, and the GnRH signaling pathway [66,67]. Furthermore, it has previously been shown that *ATF3* can directly bind to the promoter of *EGR1* to regulate its expression [68]. During calcification of eggshells, substantial quantities of Ca and bicarbonate ions are required, alongside Na and K ions, to maintain acid–base balance [1]. In the PE group, we observed downregulation of 15 metal ion binding-related genes including *ADCY6*, *ATP1A1*, *EGR1*, and *FAM20C*, which could have disrupted normal shell calcification [40]. It is noteworthy that these ion pump and metal ion-binding genes are mostly upregulated in fibroblasts of the NE group compared to those of the PE group. This suggests that in the latter, fibroblast functionality is impaired or suppressed, which affects ion transport capacity and thus impacts the normal calcification of the eggshell. We found that genes closely associated with immune and inflammatory processes, such as *FTH1*, *GPC1*, *IL15*, *TGFB2*, *B2M*, and *WNT5A*, were upregulated in the PE group. These genes are regulated by multiple TFs, including *ATF3*, *EGR1*, *FOS*, and *JUN*, even though these TFs were downregulated in the PE group. Jia et al. found that inflammatory states affect Ca transport and ECM protein synthesis in the chicken uterus [10]. Therefore, we hypothesize that in the PE group, inflammation disrupted the secretion of ions and ECM proteins, thereby affecting normal Ca deposition and eggshell ultrastructure formation, ultimately causing the production of pimpled eggs.

However, this study still has certain limitations. The findings obtained are based solely on bioinformatics analysis and have not been confirmed through molecular experiments, which will be the focus of our future work. Although three hens may be sufficient for preliminary statistical analysis and meet the minimum experimental requirements, the small sample size may affect the reliability, reproducibility, and representativeness of the results from an experimental technical perspective. Additionally, the scATAC-seq data for each group came from only one hen, lacking biological replicates, which introduces a degree of randomness to the results. Therefore, the scATAC-seq atlas constructed in this study has limited reliability and is exploratory in nature.

## 4. Materials and Methods

### 4.1. Animals

In JinQiu Agriculture and Animal Husbandry Co., Ltd., located in Tai’an City, China, a breeding facility for Luhua chickens, a group of 450-day-old healthy egg-laying hens were reared in a cage system, with free access to feed and water. Eggshell quality was assessed over a period of 4 weeks, during which we selected 3 hens producing normal eggs (NE group) and 3 hens producing pimpled eggs (PE group) for our experiments. All the chickens used in the experiment had no genetic relationship with each other. The chickens were euthanized 7–8 h after egg laying and upper uterine tissues collected and preserved in cell preservation solution for scRNA-seq. In addition, samples from one hen per group were rapidly frozen in liquid nitrogen for scATAC-seq. All experimental procedures were conducted according to the approved protocol from the College of Animal Science and Technology, Shandong Agricultural University, China (Approval No. SDAUA-2018-018).

### 4.2. Single-Cell Sample Preparation and Sequencing

Three uterine tissue samples from each group were placed individually into 160 μL of 0.25% trypsin in Eppendorf tubes and kept on ice. The samples were incubated at 37 °C for 20 min with constant shaking. The digestion was stopped by adding 20 μL or 40 μL of fetal calf serum, and then passed through a 35 μm cell strainer and collected into 1.5 mL Eppendorf tubes. Subsequently, the cells were washed with 800 μL of PBS containing 10% fetal calf serum, and the cell concentration was determined using the Moxi Z Cell Counter (Orflo Technologies, Kirkland, WA, USA). After centrifugation, the cells were resuspended at a concentration of 1000 cells/mL. Cell suspensions were loaded into chromium microfluidic chips with 3′ chemistry and barcoded using a 10× Chromium Controller (10× Genomics, Pleasanton, CA, USA). RNA extracted from the barcoded cells was then subjected to reverse transcription. Sequencing libraries were prepared using reagents from a Chromium Single Cell 3′ v3 reagent kit (10× Genomics), following the manufacturer’s instructions. Sequencing was conducted on an Illumina HiSeq 2000 platform (Illumina Inc., San Diego, CA, USA) according to the company’s recommended procedures.

### 4.3. scRNA-Seq Data Analysis

Raw reads were demultiplexed and mapped to the chicken reference genome (refg7b) using the 10× Genomics Cell Ranger pipeline (https://support.10xgenomics.com/single-cell-gene-expression/software/pipelines/latest/what-is-cell-ranger (accessed on 25 March 2024)) with default parameters. Single-cell analyses were conducted using Cell Ranger (v6.0.2) and Seurat (v4.0.5) [69], unless otherwise specified. Briefly, we used unique molecular identifiers to count gene expression for each gene and cell barcode filtered by Cell Ranger, generating digital expression matrices. Cellranger count takes FASTQ files performs alignment, filtering, barcode counting, and UMI counting. The -nosecondary option can be added to skip secondary analysis of the feature-barcode matrix (dimensionality reduction, clustering, and visualization). To identify and remove doublets, we used DoubletFinder (v2.0.3) and employed the paramSweep (PCs = 1:15, sct = TRUE) function along with the find.pK parameter to determine the optimal pK value [70]. Cells were then filtered according to Seurat criteria: genes expressed in more than 3 cells were included, and each cell was required to express 200 to 4500 genes. Cells having > 25% of reads from mitochondrial genes were excluded. Since cells tend to cluster by individual samples, the R package Harmony (v0.1.0) was used to correct for batch effects that may arise from sequencing order between samples [71]. Data normalization, dimensionality reduction, clustering, and differential expression analysis were conducted with Seurat. Specifically, we normalized the expression data using normalization method = LogNormalize and selected the top 2000 variable genes using FindVariableFeatures with the vst method. Cell cycle effects on clustering and dimensionality reduction were mitigated with CellCycleScoring and ScaleData. Principal component analysis (PCA) was performed on these 2000 variable genes using RunPCA. Cells were clustered using FindNeighbors and FindClusters (resolution = 0.5), followed by visualization of the two-dimensional clustering results with RunUMAP. We used FindMarkers to identify marker genes for each cluster, requiring that the genes were expressed in at least 40% of cells within at least one cluster, with a log fold change > 0.6, and an adjusted *p*-value (padj) < 0.05 [72].

For differential analysis between the PE and NE groups, FindMarkers was also used with the criteria mentioned above. Cell clusters were then annotated manually using known markers reported in the literature, in conjunction with the PanglaoDB website (https://panglaodb.se/index.html (accessed on 25 March 2024)) and CellMarker website (http://xteam.xbio.top/CellMarker/ (accessed on 25 March 2024)). Clusters expressing identical marker genes were merged. Finally, we performed enrichment analysis of upregulated genes to validate the accuracy of our cell type annotations. Gene Ontology (GO) and Kyoto Encyclopedia of Genes and Genomes (KEGG) enrichment analyses were performed on the DAVID website (https://david.ncifcrf.gov (accessed on 25 March 2024)).

### 4.4. Inference of Transcription Factor Regulatory Networks

To predict TF regulatory networks within cells, the R package SCENIC (v1.1.2-2) was used for analyzing scRNA-seq data [73]. Briefly, we first constructed co-expression networks with GENIE3 (v1.4.3) and then used RcisTarget (v1.2.1) for motif enrichment and target gene prediction. Next, regulatory activity was scored using AUCell (v1.4.1), followed by the calculation of regulatory specificity scores (RSSs) for each regulator within each cell type. Target genes with a positive correlation lower than 0.03 in each TF module were excluded from subsequent analysis. Regulators with high RSS were defined as key regulators. Finally, a TF–target gene–GO regulatory network was constructed and visualized using Cytoscape (v3.10).

### 4.5. Analysis of Cell–Cell Communication

The R package CellChat (v1.1.3) was used according to the authors’ instructions to identify potential cell–cell interactions among different cell clusters in the chicken uterus [74]. Due to the incomplete annotation of receptor–ligand pairs in the chicken reference genome, we used homologous human genes for further analysis. However, due to evolutionary differences between species, this approach has certain limitations. In summary, a CellChat object was first instantiated using “createCellChat”. Then, overexpressed ligands or receptors within cells and their interactions were identified using “identifyOverExpressedGenes” and “identifyOverExpressedInteractions”. Next, communication probabilities were inferred using “computeCommunProb” and “computeCommunProbPathway”, followed by the computation of an integrated cell communication network using “aggregateNet”. Finally, the results of cell communication were visualized using “netVisual_circle”.

### 4.6. Preparation and Sequencing of scATAC-Seq Libraries

The nuclei suspension was placed into a Chromium Next GEM Chip H along with 10× reagents and was barcoded using a 10× Chromium Controller (10× Genomics, Pleasanton, CA, USA). DNA fragments from these barcoded cells were then amplified and sequencing libraries prepared using the Chromium NextGEM Single Cell ATAC Reagent Kit v1.1 (10× Genomics, Pleasanton, CA, USA), following the manufacturer’s protocol. The resulting scATAC-seq libraries were pooled and sequenced on an Illumina NovaSeq PE50 platform (Illumina Inc., San Diego, CA, USA) with 2 × 50 paired-end kits at Novogene.

### 4.7. scATAC-Seq Preprocessing

Raw sequencing data were converted to fastq format with cellranger-atac mkfastq and quality assessment and filtering of the reads were conducted using fastp. Filtered reads were then aligned to the chicken reference genome (refg7b) and quantified with cellranger-atac count (10× Genomics, Pleasanton, CA, USA) using the default parameters. Cell Ranger ATAC utilizes negative binomial generalized linear models and Wald tests to identify the accessibility of TF binding sites specific to each cell cluster. Cell Ranger ATAC was used for peak calling and cell calling to identify accessible chromatin regions and authentic cell barcodes. Next, peak annotation was performed in Cell Ranger ATAC using “bedtools closest-D = b”. Distal peaks were required to be located within 100 kb upstream or downstream of transcription termination sites. Genomic annotations of peaks were generated with ChIPseeker (v1.5.1). Clustering analysis and differential accessibility were performed using Signac (v1.3.0) on the peak-by-cell data matrix. A stringent quality control (QC) procedure was first applied to the dataset to exclude low-quality cells (peak region fragments > 2500, peak region fragments < 25,000, reads in peaks > 15%, blacklist ratio < 0.001, nucleosome signal < 4, and mitochondrial gene ratio < 0.25). We performed separate analyses for each sample, so batch correction was not applied. Doublet cells were then eliminated using ArchR (v1.0.1). Subsequently, we used Signac (v1.3.0) for term frequency–inverse document frequency normalization, focusing on the top 25% of variable features for latent semantic indexing (LSI) analysis. The best clustering results were obtained when using the first 30 principal components, which were then applied to uniform manifold approximation and projection (UMAP) for dimensionality reduction. After UMAP, a shared nearest neighbor (SNN) graph facilitated cell clustering using Seurat’s Louvain algorithm. Differential accessibility peaks for each cluster were identified using Seurat’s default FindAllMarkers function. Pseudobulk profiles, highlighting aggregated fragment stacks at specific genomic regions in each cluster, were generated using Signac. Finally, we inferred gene expression levels using GeneActivity.

### 4.8. Identification of Cell Types in scATAC-Seq Data Using scRNA-Seq

The GeneActivity function from the Signac package (v1.3.0) was first used to generate a gene activity matrix. Gene coordinates were then extracted and extended to include a 2 kb upstream region. These steps are automatically handled by the FeatureMatrix and GeneActivity functions [75]. The data were then normalized using NormalizeData and ScaleData to normalize gene activities, and anchors between the scRNA-seq and scATAC-seq datasets were identified using FindTransferAnchors, with the reduction parameter set to “CCA” and the remaining parameters set to their default values. After anchor points were identified, annotations were transferred from the scRNA-seq dataset to cells in the scATAC-seq dataset using TransferData. To validate the accuracy of the scATAC-seq data annotations, the accessibility of marker gene motif sequences within the respective cell types was examined.

## 5. Conclusions

We present the first combined scRNA-seq and scATAC-seq data of chicken uterine tissue, revealing the presence of ionocytes. Our research not only provides crucial insights into cell type-specific gene regulatory networks and mechanisms of eggshell formation in the chicken uterus, but also identifies key genes involved in the formation of pimpled eggs. These findings have significant potential for advancing our understanding of genetic regulatory networks associated with egg production in chickens.

## Figures and Tables

**Figure 1 ijms-25-13431-f001:**
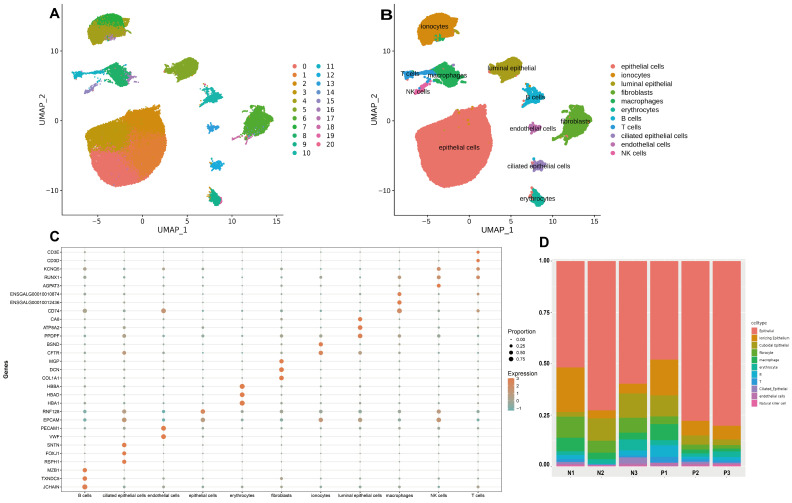
Single-cell transcriptome analysis and clustering identification of chicken uterine cells. (**A**) Unsupervised clustering revealed 21 distinct transcriptional cell clusters, which were visualized using a UMAP plot. Each point represents an individual cell, with colors indicating cluster assignments. (**B**) A UMAP plot was utilized to visualize 11 uterine cell types. Each point represents an individual cell and is color-coded according to its cell type. (**C**) The dot plot illustrates the distinct expression patterns of canonical marker genes across 11 cell populations. (**D**) Bar plot showing the percentage of different cell types within each of the 6 samples.

**Figure 2 ijms-25-13431-f002:**
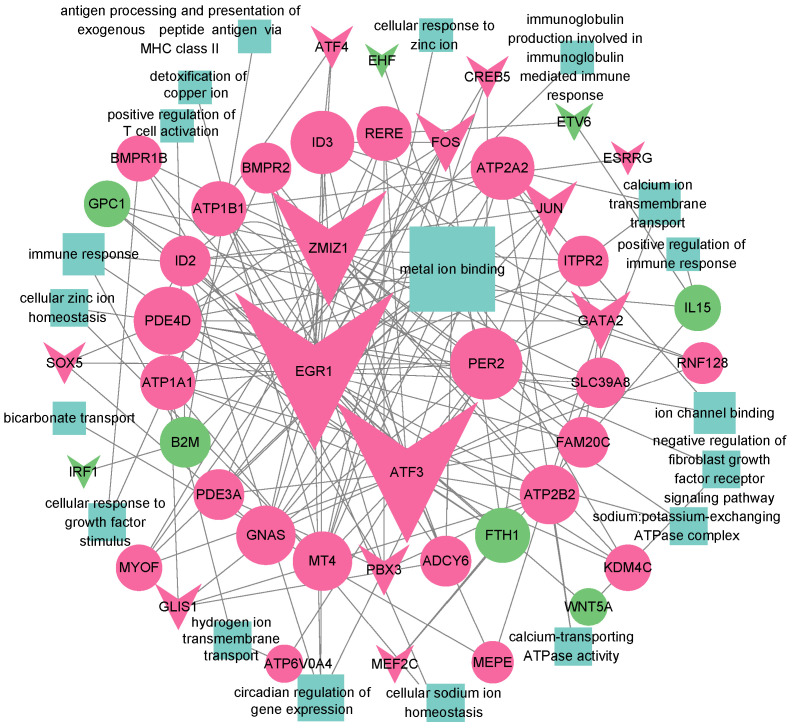
Regulatory network of DE transcription factors (TFs) and their DE target genes in chicken uterine tissue, along with GO terms. In the diagram, inverted triangles represent transcription factors (TFs), circles denote target genes, and rectangles indicate GO terms. Red highlights TFs and target genes that are upregulated in the NE group, while green highlights those upregulated in the PE group. Blue represents GO terms. The size of the shapes corresponds to the degree of involvement, with larger shapes indicating greater participation in regulatory networks.

**Figure 3 ijms-25-13431-f003:**
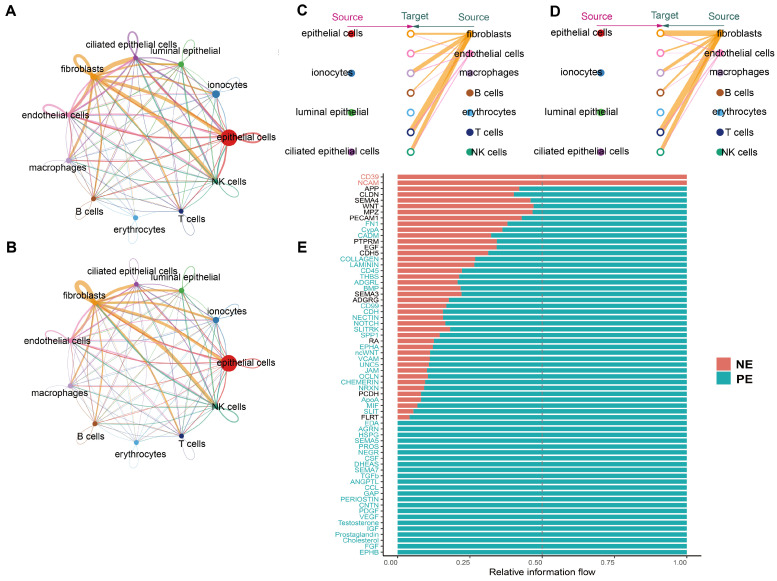
scRNA-seq reveals significant changes in cell communication between the NE and PE groups. (**A**) Number of interactions between all cells in the NE group. Thicker links indicate a higher number of interactions. (**B**) Number of interactions between all cells in the PE group. Thicker links indicate a higher number of interactions. (**C**) Cell–cell communication chord diagram for the COLLAGEN signaling pathway in the NE group, with thicker links indicating stronger interactions between cells. (**D**) Cell–cell communication chord diagram for the COLLAGEN signaling pathway in the PE group, with thicker links indicating stronger interactions between cells. (**E**) Relative signaling pathway diagram showing the pathways identified in the NE and PE groups. Larger pathways in the NE group are depicted in cyan, while those in the PE group are depicted in red. Black indicates pathways with no significant difference.

**Figure 4 ijms-25-13431-f004:**
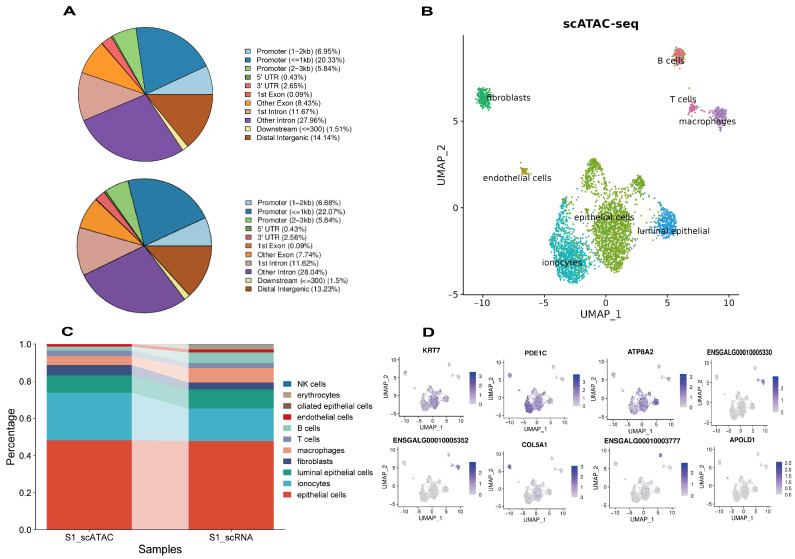
Single-Cell Chromatin Accessibility Analysis of Chicken Uterine Tissue. (**A**) Annotation and statistics of the distribution of peaks in different genomic functional regions (such as promoters, 5′UTR, 3′UTR, exons, introns, downstream regions, and intergenic regions) for the N1 sample (**top**) and P1 sample (**bottom**). (**B**) UMAP plot of the scATAC-seq dataset for sample P1, with cell type assignments based on scRNA-seq data. (**C**) Histogram of cell frequency distributions for scRNA and scATAC data in the P1 sample. (**D**) Feature plot of inferred marker gene activities, including epithelial cells (*KRT7*), ionocytes (*PDE1C*), luminal epithelial cells (*ATP8A2*), macrophages (*ENSGALG00010005330*), T cells (*ENSGALG00010005352*), fibroblasts (*COL5A1*), B cells (*ENSGALG00010003777*), and endothelial cells (*APOLD1*).

**Figure 5 ijms-25-13431-f005:**
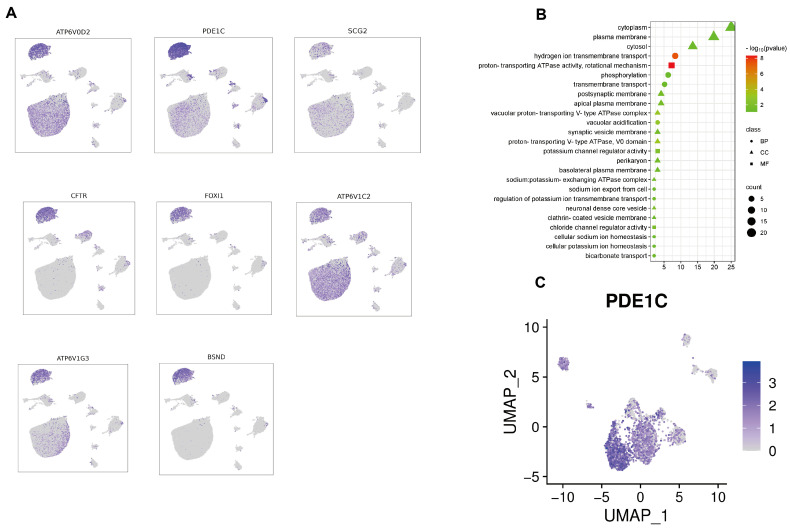
Identification of Ionocyte through marker genes and enrichment analysis. (**A**) UMAP plot showing the expression patterns of ionocyte marker genes (from the literature) in scRNA-seq data. (**B**) Enriched GO terms for genes upregulated in Ionocytes, as identified in scRNA-seq data. (**C**) UMAP visualization of PDE1C, a marker gene for ionocytes identified in the literature, showing inferred gene activity within ionocyte clusters from scATAC-seq data.

## Data Availability

All data supporting our findings are included in the manuscript. The scRNA and scATAC-seq data for this study can be found in the NCBI Sequence Read Archive (SRA) under Bioproject PRJNA19959096.

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
