# Peer review of "Integrating Single-Cell RNA-Seq and ATAC-Seq Analysis Reveals Uterine Cell Heterogeneity and Regulatory Networks Linked to Pimpled Eggs in Chickens"

_ijms, 2024, doi:10.3390/ijms252413431_

Round 1
Reviewer 1 Report
Comments and Suggestions for Authors
This study uses single-cell RNA sequencing (scRNA-seq) and single-cell ATAC sequencing (scATAC-seq) to explore the molecular pathways underlying the development of pimpled eggs, which are characterized by inadequate eggshells. The article maps 12 cell types seen in chicken uterine tissues, identifies fibroblasts as being essential for cell communication, and finds that chicks who produce pimpled eggs have increased immunological signaling. The study connected inflammation to altered ion transport gene expression and disturbed circadian rhythms, both of which have a detrimental effect on shell quality. The study's discovery of uncommon uterine ionocytes that control ion transport was interesting. These ionocytes had not been seen in any bird species before. The findings have implications for enhancing egg quality and offer new insights into the biology of avian reproduction and eggshell creation.
Even though the article appears to present interesting data at first glance, it requires further clarification and explanation of a few areas.
Although the assertion that ionocytes are "a rare cell type" found in the chicken uterus is noteworthy, further context or confirmation are needed for this discovery. Ionocytes have been observed in other systems (like fish and amphibians), which is intriguing, but this does not show that they are involved in shell production or that birds are the only animals that use them.
The introduction shifts from talking about pimpled eggs to how ageing affects uterine function, but it doesn't make a clear connection between the two topics. Although it is unclear if the study is focused on laying disparities across chickens of the same age or on age-related changes, the narrative suggests that age is a primary cause of poor egg quality.
The assertion that "the amount of pimpled eggs... increases with hens’ age" is used to reinforce the idea that eggshell defects are related to hen age. This argument is contested by the fact that the study itself employs hens of the same age. Additionally, it makes the unsupported assumption that bird and mammal systems are identical.
For transcriptome research, a sample size of three hens per group for scRNA-seq and one per group for scATAC-seq is insufficient. The possible restrictions on generalizability that this may present should be acknowledged by the authors. Each group's scATAC-seq data came from a single hen. Chromatin accessibility results are less reliable when biological replicates are not present.
According to the text, batch effects were corrected using Harmony. It does not, however, specify the type of batch effects that were noted or how they were experimentally addressed.
Because of evolutionary differences, the CellChat analysis's reliance on human homologs for chicken ligand-receptor interactions may introduce errors. This restriction has to be stated more clearly.
Eleven cell types are identified in the article, although several terminology are defined differently in the abstract and the text. Furthermore, it is unclear if the gene markers employed for cluster identification are inferred from other species or are exclusive to chicken tissues.
Although the discovery of ionocytes is new, there is insufficient data in the study to establish a direct correlation between ionocyte function and pimpled eggs or eggshell quality. This finding is speculative in the absence of functional validation.
The robustness of the results is diminished by the limited sample size for scRNA-seq and the absence of replicates for scATAC-seq, despite the presentation of criteria for differential expression analysis (e.g., log fold change, modified p-value parameters).
The analogy to studies on the uterus in humans and mammals seems unrelated. These comparisons need more convincing explanations or more seamless narrative integration because reproductive physiology differs greatly between species.
Ionocytes in chicken uteruses may reflect evolutionary differences, according to the discussion, although no thorough evolutionary analysis is done to support this assertion.
Tools such as Cell Ranger may have different "default parameters" depending on the version and process. Reproducibility might be improved by explicitly stating the software version and parameter details (such as any extra filtration thresholds).
Peak annotation was carried out in Cell Ranger ATAC using "bedtools closest -D=b," according to the article. Bedtools are not a native feature of Cell Ranger. This might be a mistake or indicate post-Cell Ranger processing, which isn't explained.
Even though rigorous, the quality control requirements (such as "minimum 200 fragments per cell") are not typical for scATAC-seq. This can result in a significant loss of data. The requirement that no more than 95% of cells surpass this cutoff also looks arbitrary and isn't a common practice.
Only the 2-kb upstream regions and gene bodies are taken into account in the GeneActivity measurement. Distal enhancers, which can significantly influence gene regulation and cell identity in ATAC-seq data, are not included in this.
There is no explanation of the batch correction procedure. For clarity, a tool or algorithm (such as Harmony or Seurat's integration) should be included. The batch correction procedure is not described. It would be clear to mention a tool or method (e.g., Harmony, Seurat's integration). Though Louvain clustering is frequent in Seurat, there is no explanation for why 30 LSI components are used for UMAP. This cutoff point could have a big impact on cluster resolution.
Although the SCENIC results indicate 280 important regulons, they do not specify a threshold for "significance." It's also ambiguous to say that "many interactions have been experimentally validated" without providing reference.
Changes in gene expression are attributed to physiological effects in the article (e.g., "clock gene expression downregulation promotes pimpled egg formation"). These assertions are not supported by causal evidence or direct experimental validation.
The parameters (such as the number of dimensions) utilized in CCA are not given, even though FindTransferAnchors is indicated for integration.
Effective clustering was impeded by "poor results" of marker gene analysis in N1, according to the report. However, this sample was still utilized to a certain measure. This throws into question the reliability of findings based on poor data. Bias is introduced by the downstream analysis's primary focus on sample P1. There is insufficient justification given for completely disregarding N1 data.
Ion transport and immune response are impacted by the downregulation of TFs (ATF3, JUN, and FOS Regulation Network) and their targets, according to the article. Nevertheless, there is inadequate evidence of direct causality (functional experiments, for example), therefore the conclusions are conjectural.
Despite the lack of conclusive sequencing evidence, EGR1 is described as both upstream and downstream of ATF3 regulation.
There are references to the differences between the uterine tissues of chickens and mammals, but no concrete comparisons are offered to back up these claims.
One piece of evidence supporting biological relevance is the GO term enrichment analysis. However, it is challenging to objectively assess the results in the absence of precise pathways, false discovery rates (FDR), or enrichment score details.
It has been found in chicken uterine tissues that inflammation disrupts circadian rhythms, however there is no concrete experimental data to support this theory (e.g., PER2, ID2 downregulation).
Minor revision
The abstract states, “Our study is the first to construct single-cell resolution transcriptomic and chromatin accessibility maps of chicken uterine tissue.” However, no explicit literature review is presented in the introduction to confirm that similar studies have not been performed in poultry. This claim needs stronger justification or clarification.
Sometimes citations are not clear (for example, "Previous research suggested…" without mentioning which research).
The text regularly cites figures and tables from supplemental materials (such as Tables S1 and S2) without providing a critical summary of the findings in the body of the article.
Author Response
Comments 1: Although the assertion that ionocytes are "a rare cell type" found in the chicken uterus is noteworthy, further context or confirmation are needed for this discovery. Ionocytes have been observed in other systems (like fish and amphibians), which is intriguing, but this does not show that they are involved in shell production or that birds are the only animals that use them.
Response 1: Thank you for pointing this out. We completely agree with this comment. Ionocytes are primarily involved in ion regulation and transport, particularly in tissues such as the skin of aquatic organisms and the respiratory and renal tissues of mammals. While ion transport in uterine tissues is closely related to eggshell formation, this does not necessarily imply that ionocytes directly contribute to eggshell formation. In our study, cell frequency analysis showed no significant difference in ionocyte populations between the NE and PE groups, suggesting that ionocytes may not directly regulate the formation of soft-shelled eggs. We plan to further investigate the potential role of ionocytes in eggshell formation in future studies. Interestingly, to our knowledge, ionocytes have not been identified in uterine tissues in single-cell transcriptomic studies of other species. What is particularly intriguing in our study is that we have identified ionocyte populations in the uterine tissues of chickens for the first time.
Comments 2: The introduction shifts from talking about pimpled eggs to how ageing affects uterine function, but it doesn't make a clear connection between the two topics. Although it is unclear if the study is focused on laying disparities across chickens of the same age or on age-related changes, the narrative suggests that age is a primary cause of poor egg quality.
Response 2: Thank you for raising this valuable point. We have revised the manuscript to clarify the ambiguous parts, in accordance with your suggestion. Regarding the formation of soft-shelled eggs, we propose that it is primarily caused by changes in uterine function. Uterine function is influenced not only by age, but also by factors such as uterine inflammation, feed composition, and environmental conditions. Therefore, by comparing the differences in uterine tissues of hens laying different types of eggs, we have further explored the mechanisms underlying the formation of soft-shelled eggs. (page 2, lines 56-61)
Comments 3: The assertion that "the amount of pimpled eggs... increases with hens’ age" is used to reinforce the idea that eggshell defects are related to hen age. This argument is contested by the fact that the study itself employs hens of the same age. Additionally, it makes the unsupported assumption that bird and mammal systems are identical.
Response 3: Thank you for pointing this out. We agree with this comment. We have made revisions based on your advice and clarified the previous ambiguities. As animals age, the normal function of uterine tissue is impaired, leading to an increased rate of soft-shelled eggs. Our main point is that the dysfunction of uterine tissue is a key cause of pimpled egg formation. (page 2, lines 56-59)
Comments 4: For transcriptome research, a sample size of three hens per group for scRNA-seq and one per group for scATAC-seq is insufficient. The possible restrictions on generalizability that this may present should be acknowledged by the authors. Each group's scATAC-seq data came from a single hen. Chromatin accessibility results are less reliable when biological replicates are not present.
Response 4: Thank you for pointing this out. We agree with this comment. Regarding the sample size issue, we fully agree that a small sample size may limit the generalizability of the results. Therefore, we have addressed this limitation in the discussion, particularly emphasizing that the chromatin accessibility data are exploratory and lack sufficient reliability due to the small sample size.(page 16, lines 592-601)
Comments 5: According to the text, batch effects were corrected using Harmony. It does not, however, specify the type of batch effects that were noted or how they were experimentally addressed.
Response 5: We greatly appreciate your valuable feedback. In this study, all six chickens were kept under the same environmental conditions, with identical feeding protocols, sampling times, and procedures. However, there was a difference in the sequencing order of the samples, and we observed a trend where cells tended to cluster according to individual samples. To address this, we applied Harmony for batch correction. This has been explained in the revised manuscript.(page 4, lines 157-159)
Comments 6: Because of evolutionary differences, the CellChat analysis's reliance on human homologs for chicken ligand-receptor interactions may introduce errors. This restriction has to be stated more clearly.
Response 6: We greatly appreciate your valuable feedback. We understand and agree that the CellChat analysis method, which relies on human homologs to infer ligand-receptor interactions in chickens, may introduce some potential errors. Currently, due to the lack of a species-specific ligand-receptor interaction database for chickens, this method is widely used for cross-species inference. However, we also recognize that this inference strategy may not be entirely accurate. Therefore, we will more clearly address this limitation in the revised manuscript. (page 4, lines 193-195)
Comments 7: Eleven cell types are identified in the article, although several terminology are defined differently in the abstract and the text. Furthermore, it is unclear if the gene markers employed for cluster identification are inferred from other species or are exclusive to chicken tissues.
Response 7: Thank you very much for your valuable feedback. We have re-examined and standardized the terminology used in both the abstract and the main text. Since there are relatively few single-cell sequencing studies conducted on chickens, and most studies rely on gene markers from humans and mice, the gene markers we used were primarily based on relevant literature from humans and mice. Additionally, some gene markers were sourced from the PanglaoDB website (https://panglaodb.se/index.html) and CellMarker (http://xteam.xbio.top/CellMarker/). We have provided clarification on this in the revised manuscript. (page 4, lines 172-175)
Comments 8: Although the discovery of ionocytes is new, there is insufficient data in the study to establish a direct correlation between ionocyte function and pimpled eggs or eggshell quality. This finding is speculative in the absence of functional validation.
Response 8: Thank you for your valuable feedback on our study. We have not yet conducted comprehensive functional validation, so we are unable to directly confirm the causal relationship between ionocytes and eggshell quality. Therefore, in the revised manuscript, we have not associated ionocytes with eggshell quality but have only stated that this cell type has not been found in the uterine tissues of other mammals. (page 14, lines 490-491)
Comments 9: The robustness of the results is diminished by the limited sample size for scRNA-seq and the absence of replicates for scATAC-seq, despite the presentation of criteria for differential expression analysis (e.g., log fold change, modified p-value parameters).
Response 9: Thank you for your valuable feedback. We fully understand that the limited sample size and lack of replication may impact the robustness of the results. Given the sample size limitations, there are indeed some statistical power concerns. In the revised manuscript, we will further emphasize the potential impact of the small sample size on the interpretation of the results. (page 16, lines 592-601)
Comments 10: The analogy to studies on the uterus in humans and mammals seems unrelated. These comparisons need more convincing explanations or more seamless narrative integration because reproductive physiology differs greatly between species.
Response 10: Thank you for pointing this out. We agree with this comment. Since we did not perform a comparative analysis between the mammalian uterine data and the chicken single-cell data, the analogy between the two is somewhat unnecessary. Based on your feedback, we have reduced this section in the revised manuscript.
Comments 11: Ionocytes in chicken uteruses may reflect evolutionary differences, according to the discussion, although no thorough evolutionary analysis is done to support this assertion.
Response 11: Thank you for pointing this out. We agree with this comment. We found ionocytes in chicken uterine tissue, whereas they have not been identified in single-cell studies of other mammalian uterine tissues. Therefore, we speculate that this difference may be due to evolutionary divergence between the two. However, this result is purely speculative. We will consider incorporating more evolutionary biology analyses in future studies to further explore species-specific differences in uterine cell types.
Comments 12: Tools such as Cell Ranger may have different "default parameters" depending on the version and process. Reproducibility might be improved by explicitly stating the software version and parameter details (such as any extra filtration thresholds).
Response 12: Thank you for pointing this out. In the revised version, we have added detailed information about the versions of tools used, such as Cell Ranger, along with the specific parameter settings and any additional filtering thresholds. Thank you again for your suggestion, which has helped us improve the accuracy and reproducibility of our paper. (page 3, lines 149-154)
Comments 13: Peak annotation was carried out in Cell Ranger ATAC using "bedtools closest -D=b," according to the article. Bedtools are not a native feature of Cell Ranger. This might be a mistake or indicate post-Cell Ranger processing, which isn't explained.
Response 13: Thank you for pointing this out. Bedtools is not a native feature of Cell Ranger, and this was an error in our description. We will make the necessary correction in the revised manuscript. Thank you again for your thorough review and constructive suggestions.
Comments 14: Even though rigorous, the quality control requirements (such as "minimum 200 fragments per cell") are not typical for scATAC-seq. This can result in a significant loss of data. The requirement that no more than 95% of cells surpass this cutoff also looks arbitrary and isn't a common practice.
Response 14: Thank you for pointing this out. We have realized that there were inaccuracies in the description of this section, and we have now made the necessary revisions to the related content. (page 5, lines 223-227)
Comments 15: Only the 2-kb upstream regions and gene bodies are taken into account in the GeneActivity measurement. Distal enhancers, which can significantly influence gene regulation and cell identity in ATAC-seq data, are not included in this.
Response 15: Thank you very much for your valuable feedback. We sincerely apologize for any misunderstanding caused by our inaccurate description, and we have made the necessary revisions. In the GeneActivity measurement section, the method we used is automatically performed using the FeatureMatrix and GeneActivity functions in the Signac software (https://stuartlab.org/signac/articles/pbmc_vignette.html). (page 5, lines 240-247)
Comments 16: There is no explanation of the batch correction procedure. For clarity, a tool or algorithm (such as Harmony or Seurat's integration) should be included. The batch correction procedure is not described. It would be clear to mention a tool or method (e.g., Harmony, Seurat's integration). Though Louvain clustering is frequent in Seurat, there is no explanation for why 30 LSI components are used for UMAP. This cutoff point could have a big impact on cluster resolution.
Response 16: Thank you very much for your valuable feedback. Since each dataset in the scATAC-seq analysis contains only one sample per group, we did not perform batch effect correction and instead chose to analyze each sample separately. We will further clarify this point in the revised version to provide a clearer explanation of our analysis strategy. (page 5, lines 226-227)
Regarding the choice of using the first 30 LSI components for UMAP dimensionality reduction, we found during the analysis that using the first 30 principal components provided the best dimensionality reduction results. Therefore, we adopted this approach. In the revised version, we will include additional explanation to further justify the rationale for this choice. (page 5, lines 228-232)
Comments 17: Although the SCENIC results indicate 280 important regulons, they do not specify a threshold for "significance." It's also ambiguous to say that "many interactions have been experimentally validated" without providing reference.
Response 17: Thank you for pointing this out. We fully agree with your comments, and in the revised version, we have added the filtering threshold for the SCENIC results. Additionally, we recognized that the phrase 'many interactions have been experimentally validated' was not sufficiently rigorous, so we have revised it in the updated manuscript. (page 4, lines 186-187)
Comments 18: Changes in gene expression are attributed to physiological effects in the article (e.g., "clock gene expression downregulation promotes pimpled egg formation"). These assertions are not supported by causal evidence or direct experimental validation.
Response 18: Thank you very much for your valuable comments. We fully agree with your perspective and have made the corresponding revisions in the revised manuscript. (page 7, lines 302-305, page 15, lines 548-551)
Comments 19: The parameters (such as the number of dimensions) utilized in CCA are not given, even though FindTransferAnchors is indicated for integration.
Response 19: Thank you for pointing this out. It was an oversight on our part not to specify the parameters used in this analysis, and we have now added the necessary details in the revised manuscript. (page 5, lines 243-247)
Comments 20: Effective clustering was impeded by "poor results" of marker gene analysis in N1, according to the report. However, this sample was still utilized to a certain measure. This throws into question the reliability of findings based on poor data. Bias is introduced by the downstream analysis's primary focus on sample P1. There is insufficient justification given for completely disregarding N1 data.
Response 20: Thank you to the reviewer for the careful review and valuable comments on our work. The sequencing quality of the N1 sample was acceptable; however, the clustering analysis showed that almost all cells grouped together, making it impossible to perform cell grouping. Therefore, the data from the N1 sample were not used in the subsequent analysis. We have added the corresponding explanation in the revised manuscript. In the downstream analysis, we did not perform inter-group comparisons, but instead focused on whether the motifs with higher chromatin accessibility in each cell cluster of the P1 sample could annotate that cell cluster, and the role they played in specific cell clusters. (page 11, lines 418-420)
Comments 21: Ion transport and immune response are impacted by the downregulation of TFs (ATF3, JUN, and FOS Regulation Network) and their targets, according to the article. Nevertheless, there is inadequate evidence of direct causality (functional experiments, for example), therefore the conclusions are conjectural.
Response 21: Thank you for your valuable comments. We did not provide direct causal evidence to prove the relationship between ion transport, immune response, and the downregulation of TFs (ATF3, JUN, and FOS) and their targets. We only made predictions based on relevant literature and analyses from this study. We agree with your point and have clarified this in the revised manuscript. (page 8, lines 347-350)
Comments 22: Despite the lack of conclusive sequencing evidence, EGR1 is described as both upstream and downstream of ATF3 regulation.
Response 22: Thank you for your valuable comments. In our transcription factor analysis, we identified the reciprocal regulatory relationship between EGR1 and ATF3 based on SCENIC results (spearCor: 0.65, CoexWeight: 0.061). Additionally, previous literature has also discussed this regulatory interaction (DOI: 10.1042/bj20120125). We will provide further clarification on this point in the revised manuscript to enhance the reliability of the results."
Comments 23: There are references to the differences between the uterine tissues of chickens and mammals, but no concrete comparisons are offered to back up these claims.
Response 23: Thank you for your valuable comments. In the revised manuscript, we have added relevant literature to better support the differences between avian and mammalian uterine tissues. We also plan to further investigate these differences in future research by conducting a comparative analysis of single-cell atlases of avian and mammalian uterine tissues. (page 14, lines 517-520)
Comments 24: One piece of evidence supporting biological relevance is the GO term enrichment analysis. However, it is challenging to objectively assess the results in the absence of precise pathways, false discovery rates (FDR), or enrichment score details.
Response 24: Thank you for your careful review and valuable suggestions regarding our work. We fully agree with your recommendation on the GO term enrichment analysis, and in the revised manuscript, we have marked the significance of the GO terms accordingly. (such as on page 6, lines 278, 279, 281, and on page 7, lines 304, 308, etc.)
Comments 25: It has been found in chicken uterine tissues that inflammation disrupts circadian rhythms, however there is no concrete experimental data to support this theory (e.g., PER2, ID2 downregulation).
Response 25: Thank you for the valuable comments provided by the reviewer. Regarding the hypothesis on the relationship between inflammation and circadian rhythm, it is based on relevant studies from existing literature. However, due to the lack of specific experimental data, we will clarify this point in the revised manuscript, emphasizing that we are only proposing a hypothesis that inflammation in chicken uterine tissue may disrupt the circadian rhythm, rather than drawing definitive experimental conclusions. We appreciate the reviewer's suggestion and will refine this section further during the revision process.(page 14, lines 548-553)
Minor revision
Comments 26: The abstract states, “Our study is the first to construct single-cell resolution transcriptomic and chromatin accessibility maps of chicken uterine tissue.” However, no explicit literature review is presented in the introduction to confirm that similar studies have not been performed in poultry. This claim needs stronger justification or clarification.
Response 26: Thank you for your valuable suggestions. In the abstract, we stated that we "first constructed single-cell resolution transcriptomic and chromatin accessibility maps of chicken uterine tissue," primarily because there is no existing research on chicken uterine tissue at the single-cell level, nor are there relevant reviews in the literature. We understand your concern about this statement, as we were unable to obtain sufficient evidence to support this conclusion. Therefore, we have made revisions to this part in the revised manuscript. (page 1, lines 26-28)
Comments 27:Sometimes citations are not clear (for example, "Previous research suggested…" without mentioning which research).
Response 27: Thank you for your feedback. We apologize for the lack of clarity in some of the citations. In the revised manuscript, we have made corrections to the unclear citations. Thank you again for your valuable comments. (page 8, lines 364-365, page 12, lines 448-450, page 12, lines 467-468, page 14, lines 515-516, page 15, lines 543-545)
Comments 28:The text regularly cites figures and tables from supplemental materials (such as Tables S1 and S2) without providing a critical summary of the findings in the body of the article.
Response 28: We appreciate the valuable comments provided by the reviewer. In the revised manuscript, we have included summaries of all supplementary tables and figures to better highlight their key findings within the main text. Thank you again for your comments; they have been extremely helpful in improving our manuscript. (page 17, lines 644-670)
Once again, we appreciate the reviewer’s attention to our research, and we will carefully consider these suggestions during the revision process to improve the manuscript.
Reviewer 2 Report
Comments and Suggestions for Authors
Dear authors,
Here are a few suggestions to improve the manuscript presenting your research. The results themselves are interesting and sounding in the appraisal and understanding of the genetic origin and metabolic and transcriptomic cause-effect chain of obtaining eggs with shell defects.
In the introduction section, please check again the scientific literature and find out that loses on the logistic chain of egg production-delivery-retail are higher than 8-11%...
My main concern regarding the consistency of your results is the small number of individuals you have used in your experiment. Only 3 hens per group could be minimum required from a statistic, pure mathematical point of view to be able to run statistical computation. However, from an experimental technique point of view, this quite minuscule sampling pool is not enough to have representativity for millions of laying hens deployed in farms, that produce shells with integer or with abnormal shells.
Therefore, for this reason, I encourage you to reprocess the data using a wider sample base and I will recommend major revision.
I reiterate here the fact that the ideas of the research and the results could be really useful if they would come forth form a larger sampling pool, otherwise they are just good as theory.
May you have the patience and dedication in reprocessing the data and resubmitting the manuscript, because the results seem to be quite valuable for the avicultural practice.
Thank you indeed!
Author Response
Comments 1: In the introduction section, please check again the scientific literature and find out that loses on the logistic chain of egg production-delivery-retail are higher than 8-11%...
Response 1: Thank you to the reviewer for your suggestions on the introduction. Based on your feedback, we conducted further investigation and found that the actual proportion of egg product loss is between 12% and 20%. We have made the necessary changes in the revised manuscript. (page 1, lines 36-37)
Comments 2: My main concern regarding the consistency of your results is the small number of individuals you have used in your experiment. Only 3 hens per group could be minimum required from a statistic, pure mathematical point of view to be able to run statistical computation. However, from an experimental technique point of view, this quite minuscule sampling pool is not enough to have representativity for millions of laying hens deployed in farms, that produce shells with integer or with abnormal shells.
Response 2: Thank you to the reviewer for their valuable comments on our research. We understand the concern regarding the sample size. Given the high cost of single-cell transcriptome sequencing at the experimental design stage, the selection of three samples per group was based on practical considerations, taking into account the experimental conditions and available resources. With the limited budget, we aimed to meet the basic statistical requirements (3 vs. 3). However, we also acknowledge that a small sample size may have certain limitations in terms of representativeness and the broader applicability of the results. In our future studies, we will consider increasing the sample size to further enhance the reliability and representativeness of the findings. (page 16, lines 592-601)
During the sampling period, we conducted a four-week egg quality assessment on the experimental chickens. The final samples used for sequencing were selected from chickens that consistently produced normal eggs (NE group) or eggs with severe shell defects (PE group) throughout the observation period, ensuring greater reliability in the analysis.
Once again, we appreciate the reviewer’s attention to our research, and we will carefully consider these suggestions during the revision process to improve the manuscript.
Reviewer 3 Report
Comments and Suggestions for Authors
The manuscript entitled: "Integrating single-cell RNA-seq and ATAC-seq analysis reveals uterine cell heterogeneity and regulatory networks linked to pimpled eggs in chickens" presents the first single-cell RNA-seq and scATAC-seq analysis of the uterus in hens. Additionally, by comparing the transcriptomic profiles in hens laying normal eggs and pimpled, the authors try to identify the genes responsible for the appearance of this unfavorable trait. The subject of the work is important and current both from a practical point of view and from the level of basic research. Nevertheless, the work contains many methodological inconsistencies that need to be explained.
1. Only one sample per group was used for the ATAC-seq analysis - is this not too little to draw conclusions?
2. The authors claim that inflammation of the uterus may be responsible for the appearance of pimpled eggs, wouldn't we expect infiltration of immune cells in such a case?
3. What did the authors mean: line 390 "Because of poor results of the marker gene analysis of sample N1.."
4. line 125 Please provide the detail of cell and nuclei suspensions preparation
5. lines 113 were all the eggs layed by P group hens pimpled ?
6. were the animals fed the same feed
7. were the animals related?
8. There is no supplementary tables
9. Figure 1 line 241 oviduct cells?
10. Discussion: I suggest to divide the discussion to subchapters describing the most important findings of the work
Author Response
Comments 1: Only one sample per group was used for the ATAC-seq analysis - is this not too little to draw conclusions?
Response 1: We thank the reviewer for their valuable comments on our study. Regarding the use of only one sample per group for the scATAC-seq analysis, we recognize that the small sample size may affect the representativeness of the results and the reliability of the conclusions. In this study, considering the preliminary exploratory nature of the experimental design, the relatively immature scATAC-seq technology, and the high sequencing costs, we opted to use a single sample per group for initial analysis. Therefore, the current scATAC-seq results should be considered exploratory, with relatively low reliability. In the revised manuscript, we will further discuss this limitation and plan to increase the sample size in future studies to improve the reproducibility and statistical significance of these preliminary findings. (page 16, lines 592-601)
Comments 2: The authors claim that inflammation of the uterus may be responsible for the appearance of pimpled eggs, wouldn't we expect infiltration of immune cells in such a case?
Response 2: Thank you for your valuable comments on our work. We fully agree with your point, and indeed, the infiltration of immune cells is an important aspect that deserves further attention.
In our study, based on the results of differential expression analysis and enrichment analysis, we observed upregulation of genes related to inflammation in the shell-less egg group. Therefore, we proposed that uterine inflammation could be a potential cause for the formation of shell-less eggs. We greatly appreciate your suggestion regarding immune cell infiltration, which provides us with a new direction for further exploring this mechanism. We plan to include related experiments in our future research to investigate the potential role of immune cells in this process. Once again, thank you for your constructive feedback.
Comments 3: What did the authors mean: line 390 "Because of poor results of the marker gene analysis of sample N1.."
Response 3: We would like to thank the reviewer for the careful review and valuable comments on our work. In this study, the sequencing quality of the N1 sample was acceptable, but the clustering analysis showed that almost all cells clustered together as a single group, making it impossible to perform cell grouping. Therefore, we did not use the data from the N1 sample in the subsequent analysis. We greatly appreciate your feedback, and we have provided a more detailed explanation of this issue in the revised manuscript.(page 11, lines 418-420)
Comments 4: line 125 Please provide the detail of cell and nuclei suspensions preparation
Response 4: Thank you to the reviewer for the detailed review of our work. Regarding the preparation details of the cell and nuclear suspensions mentioned in line 125, we have provided additional clarification in the revised manuscript. (page 3, lines 128-135)
Comments 5: lines 113 were all the eggs layed by P group hens pimpled ?
Response 5: Thank you for pointing this out. Yes, all the eggs laid by the hens in the P group were pimpled eggs. Prior to sample collection, we first conducted a four-week egg quality assessment, recording the quality of eggs laid by each hen daily. Ultimately, we selected three hens that consistently laid smooth, normal eggs as the NE group, and three hens that consistently laid pimpled eggs as the PE group.
Comments 6: were the animals fed the same feed?
Response 6: Thank you very much for your feedback. These experimental chickens were all raised in the same poultry house under uniform management, with the same environmental conditions and feed. They had free access to water, and the sampling time was the same for all, in order to minimize batch effects between samples.
Comments 7: were the animals related?
Response 7: Thank you for your valuable feedback. In our experiment, the chickens used have no genetic relationship with each other. We will clearly specify this in the revised manuscript to avoid any potential confusion. Once again, we appreciate your careful review and feedback. (page 3, lines 120)
Comments 8: There is no supplementary tables
Response 8: Thank you to the reviewer for pointing out this issue. We have provided all the supplementary tables and figures in the revised manuscript and supplementary files, ensuring that all data and information are clearly presented. We hope this will help improve the manuscript. Thank you once again for your thorough review.
Comments 9: Figure 1 line 241 oviduct cells?
Response 9: Thank you very much for your valuable feedback. We recognize that the description here was ambiguous and have made revisions in the manuscript accordingly. We appreciate your careful review. (page 6, lines 265)
Comments 10: Discussion: I suggest to divide the discussion to subchapters describing the most important findings of the work
Response 10: Thank you for your valuable suggestions. We fully agree with your comments. In the revised manuscript, we have made some modifications to the discussion section to enhance the structure and readability of the article.
Once again, we appreciate the reviewer’s attention to our research, and we will carefully consider these suggestions during the revision process to improve the manuscript.
Round 2
Reviewer 1 Report
Comments and Suggestions for Authors
In accordance with the instructions and suggestions given, the article has undergone significant revisions. The detailed answers are also persuasive. I have nothing more to say. Good luck.
Reviewer 2 Report
Comments and Suggestions for Authors
Dear authors,
Thank you for replying to the suggestions I have made.
I understand your explanation related to the sampling size, based on the bidget that would really increase if you have got more subjects for sampling.
However, despite the very accurate analysis you have run on the samples you have taken, I still consider the sampling pool is by far too small and I will leave the final decision for publishing to the academic editor that is able to ask the opinion of the other reviewers about this fact.
Thank you indeed!
Reviewer 3 Report
Comments and Suggestions for Authors
The authors have slightly improved the manuscript, but I still have the impression that the results of the analyses have not been analyzed thoroughly enough, e.g. the authors emphasize the enrichment of DEGs in processes related to immunology, meanwhile none of them obtained FDR<0.05, in contrast to other processes. Moreover, the authors do not indicate in the conclusions what the main genes and processes responsible for the formation of pimpled eggs are. I also wonder if the mapping ratio around 60% is acceptable for this type of analysis